# A Two-Stage Real-Time Optimization Model of Arterial Signal Coordination Based on Reverse Causal-Effect Modeling Approach

Binbin Hao [ID], Bin Lv *, Qixiang Chen and Xianlin Li

School of Traffic and Transportation, Lanzhou Jiaotong University, Lanzhou 730070, China;
hbb@lzjtu.edu.cn (B.H.); chqixiang@sina.cn (Q.C.); 13220004@stu.lzjtu.edu.cn (X.L.)
* Correspondence: jdlbxx@mail.lzjtu.cn

**Abstract:** The arterial signal coordination is an effective method to improve traffic operational efficiency and reduce vehicle delay. In this paper, a two-stage arterial signal coordination model under dynamic traffic demands is established, and the signal timing and offset are adjusted according to the dynamic traffic demands. The objective is to minimize the expected intersection delay and the overflow of the coordinated direction. In the first stage, a calculation model for intersection signal timing based on phase clearing reliability is proposed by the reverse causal-effect modeling approach, which can calculate the signal timing of each intersection in real time. In the second stage, an offset calculation model is established to achieve the goal of minimizing delay in the coordinated direction, which can calculate the offset of trunk coordination in real time. The concept of phase clearance reliability is introduced in the model, which can dynamically adjust the balance between the coordinated phase and the non-coordinated phase, thus taking the overall control efficiency of intersections into account. We then develop an algorithm to solve the problem and then apply the model and the solution algorithm to an arterial road with three intersections to investigate and compare its performance with the Allsop's method and the Webster's method. A comparison between the proposed coordinated two-stage logic and a coordinated actuated logic is also conducted in the case study to show the advantages and disadvantages.

**Keywords:** traffic engineering; signal coordination; delay; overflow

## 1. Introduction

Urban arterial roads are distributed with major traffic flows, and the arterial signal coordination can reduce vehicle delays, stopping times, and fuel consumption. Hence, it is important for the whole city to improve traffic operational efficiency. Generally, the optimization methods of arterial signal coordination can be summarized into two categories: the maximizing bandwidth method and the minimizing performance indicator method, such as vehicle delay, stopping times, queue length, etc.

The maximizing bandwidth method takes the maximum bandwidth of arterial signal as the optimization goal. Little established the Maxband model in 1966 with phase sequence, offset, and signal period as constraints and the maximum bandwidth as the optimization objective [1]. Subsequent scholars have improved the Maxband model. Gartner et al. established a variable bandwidth Multiband model based on the Maxband model, considering the traffic flow, traffic conditions, and different bandwidth requirements, so that each phase could receive a separate weighted bandwidth [2]. Zhang et al. proposed the AM-band model, which is different from the Maxband and Multiband models [3]. It took the asymmetric and unequal constraints into consideration. Yu et al. improved the Multiband model, added the constraint condition of bandwidth ratio, and established the queue discharge time model [4]. Zhang et al. took the maximization of two-way bandwidth as the optimization goal and designed the ripple changes to improve the two-way bandwidth without changing the green wave speed [5]. Yu et al. proposed to optimize the

subsystems partition method and signal coordination scheme considering breakpoint cost based on the classical signal coordination scheme model [6].

The minimizing performance indicator method takes the control performance indexes of arterial coordinated intersections as the optimization goal. Clayton et al. proposed a signal timing optimization model with minimum delay based on the analysis of the functional relationship between vehicle delay and green time [7]. Webster, on the basis of the Clayton model, assumed that the arrival rate, departure rate, and capacity remained stable within the time interval and proposed the calculation method of signal timing parameters and the Webster delay model [8]. Hillier et al. established the functional relationship between the total delay time of vehicles and offset by analyzing the dissipation process of the queue at adjacent intersections and, on this basis, established the arterial signal coordination model with minimum delay as the optimization objective [9]. Lieberman et al. also took the minimum delay as the objective function and established the SIGOP system of arterial signal timing optimization with the dynamic programming method [10]. Benekohal et al. analyzed the difference in vehicle arrival rates at downstream intersections and established a delay model using the arrival-based method to realize multi-section and multi-mode arterial signal coordination [11]. Based on the heuristic search method, Shenoda et al. proposed an adaptive signal control model aimed at minimum delay [12]. By analyzing the relationship between overflow queue and delay, Ma et al. established a multi-stage stochastic program for arterial signal coordination—with the minimum delay and overflow queue in the coordinated direction as the optimization objective—and proposed a gradient descent solving algorithm based on phase clearing reliability [13]. Based on shockwave theory, Wang et al. analyzed the relationship between the delay in the coordinated direction and offset and established an oversaturated arterial signal coordination model [14].

The coordinated actuated control method is also the focus of scholars' research. The coordinated actuated control method can effectively reduce traffic flow delay and travel time under the condition of traffic flow fluctuation [15,16]. Yin et al. conducted a separate optimization study on the period, offset, and green split of the coordinated actuated control model [17]. Using the convenience of the cellular automata model, Zhang et al. established a mixed integer nonlinear programming model for coordinated actuated signal [18]. Cesme et al. proposed a new adaptive control model based on single intersection actuated signal control and added additional rules to realize arterial signal coordination [19]. He et al. realized the arterial coordinated actuated signal by adding virtual requests to the actuated signal control model [20].

To summarize, the existing literature has proposed many optimization strategies for arterial signal coordination, but there is still room for improvement: (1) The coordinated form of the green wave signal is simple and easy to achieve, but its optimization objective is relatively simple—it fails to quantitatively reflect the actual operational state of traffic flow and suffers from deficiencies in the actual usage process. (2) Coordinated actuated signal can better adapt to fluctuating traffic flow, and it is better suited for small traffic flow. However, when the traffic flow is large, the green time will be opened early or interrupted prematurely, and the green time will be inefficient. (3) The arterial signal coordination method with the objective of optimizing the performance indicator (minimizing vehicle delay, queue length, etc.) can overcome the shortcomings of green wave signal coordination, and its evaluation of arterial coordination effect is more specific. However, the relationship between offset and traffic delay has not been explicitly revealed in the existing literature, and most of the relevant models for signal timing are nonlinear. Zhai analyzed the relationship between delay and offset in the arterial coordinated direction and established an offset optimization model with the minimum delay in the arterial coordinated direction as the optimization objective [21]. The model clearly reveals the relationship between the offset, delay, signal timing, saturation flow rate, and arrival rate in the arterial coordinated direction, and the offset optimization is based on the signal timing scheme. However, the basis for optimizing offset in this model is signal timing, and the model failed to achieve

synchronous optimization of the offset and timing scheme. Liu et al. established a dynamic linear programming model for signal timing at a single intersection using the reverse causal-effect modeling approach, with the objective of minimizing the total delay [22]. This model is a linear programming model that can be quickly solved for signal timing at a single intersection. However, this model can only be used to solve signal timing at a single intersection and does not achieve arterial signal coordination. Based on Zhai's model and Liu's model, this paper establishes a two-stage arterial signal coordination model. In the first stage, a calculation model for intersection signal timing based on phase clearing reliability is proposed by the reverse causal-effect modeling approach, which can calculate the signal timing of each intersection in real time. In the second stage, an offset calculation model is established to achieve the goal of minimizing delay in the coordinated direction, which can calculate the offset of trunk coordination in real time. This model can be used for real-time coordinated control of arterial signals. Additionally, the efficiency control of the coordinated phase and the non-coordinated phase can be taken into account by flexibly adjusting the phase clearance reliability.

## 2. Model Formulation

### 2.1. Offset Model for Two-Way Arterial Signal Coordination

The offset is an important parameter of arterial signal coordination. Reasonable setting of offset can ensure the smooth operation of arterial traffic flow, reduce the delay and queue length at intersections, and improve the traffic efficiency. The nomenclature list of this paper are shown in Table 1.

**Table 1.** Nomenclature list.

| | |
|---|---|
| $n$ | The set of intersection, indexed by $i$ |
| $k$ | The number of cycles |
| $g_p^i$ | The green time of intersection $i$ in phase $p$ (s) |
| $r_p^i$ | The red time of intersection $i$ in phase $p$ (s) |
| $o^{i,i+1}$ | The offset between the adjacent intersection $i$ and $i + 1$ (s) |
| $\eta_p^i$ | The green time ratio in phase $p$ of intersection $i$ |
| $P$ | The set of signal phases, indexed by $p$ |
| $C$ | The length of a signal cycle (s) |
| $M^p$ | The set of allowable traffic streams in phase $p$, indexed by $m$ |
| $\lambda_{p_m}^i$ | The arrival rate for traffic stream m in phase $p$ of intersection $i$ (veh/h) |
| $s_p^i$ | The average saturation outflow rate of intersection $i$ in phase $p$ (veh/h) |
| $\mu_{p_m}^i$ | The outflow rate for traffic stream m in phase $p$ of intersection $i$ (veh/h) |
| $v^{i,i+1}$ | The average speed of vehicles from intersection $i$ to intersection $i + 1$ (m/s) |
| $L^{i,i+1}$ | The distance from intersection $i$ to intersection $i + 1$ (m) |
| $t^{i,i+1}$ | The time difference between the time at the start of green at intersection $i$ and the time when the first car at intersection $i$ arrives at intersection $i + 1$ (s) |
| $Q_{p_m}^i$ | The remaining queue for traffic stream m in phase $p$ of intersection $i$ (veh) |

The vehicle arrival rate at the intersection is a time-varying value that can be regarded as a function of time variation. For a two-way arterial signal coordination, the traffic flow from $i$ to $i + 1$ will exhibit discrete and time-varying characteristics. The traffic flow arriving at intersection $i + 1$ is irrelevant to its signal timing, but the departing traffic flow is related. The most direct correlation between departing traffic flow and signal timing is the offset $o^{i,i+1}$, and the vehicle delay in the coordinated direction is also directly related to the offset $o^{i,i+1}$ [21].

The average travel time of upstream and downstream traffic at any adjacent intersection $T_u^{i,i+1}$ and $T_d^{i+1,i}$ can be expressed as

$$T_u^{i,i+1} = \frac{L_u^{i,i+1}}{v_u^{i,i+1}} \tag{1}$$

$$T_d^{i+1,i} = \frac{L_d^{i+1,i}}{v_d^{i+1,i}} \tag{2}$$

Due to the periodicity of signal timing and vehicle arrival rates, vehicle delay in any cycle is representative of the whole. Therefore, $t^{i,i+1}$ and $t^{i+1,i}$ can be described as

$$t^{i,i+1} = T_u^{i,i+1} \bmod(C) \tag{3}$$

$$t^{i+1,i} = T_d^{i+1,i} \bmod(C) \tag{4}$$

Obviously, $t^{i,i+1} \in [0, C)$, $t^{i+1,i} \in [0, C)$.

Because of the periodicity of the traffic flow, the interrelation between signal timing, offset, and delay can be analyzed in one cycle and then extended to the whole period.

$$\lambda_{p_{\text{coor}}}^{i+1}(t) = \begin{cases} 0 & ,t < t^{i,i+1} \\ \lambda_{p_{\text{coor}}}^{i+1}(t), & t^{i,i+1} \leq t \leq t^{i,i+1} + T_u^{i+1} \\ 0 & ,t > t^{i,i+1} + T_u^{i+1} \end{cases} \tag{5}$$

$$\lambda_{p_{\text{coor}}}^{i}(t) = \begin{cases} 0 & ,t < o^{i,i+1} + t^{i+1,i} \\ \lambda_{p_{\text{coor}}}^{i}(t), & o^{i,i+1} + t^{i+1,i} \leq t \leq o^{i,i+1} + t^{i+1,i} + T_d^{i} \\ 0 & ,t > o^{i,i+1} + t^{i+1,i} + T_d^{i} \end{cases} \tag{6}$$

$$\begin{cases} g_{p_{\text{coor}}}^{i} < T_d^{i} \leq C + r_{p_{\text{coor}}}^{i} \\ g_{p_{\text{coor}}}^{i+1} < T_u^{i+1} \leq C + r_{p_{\text{coor}}}^{i+1} \end{cases} \tag{7}$$

This paper assumes that the traffic flow in the coordinated direction is undersaturated, that is, the traffic flow in the coordinated direction satisfies Formulas (5) and (6), and the duration of the arrival rate of the traffic flow satisfies Formula (7). Figure 1 shows that the vehicle delay at intersections $i$ and $i + 1$ within one cycle includes the following two aspects: (1) delay before and during the green time, (2) delay during the red time.

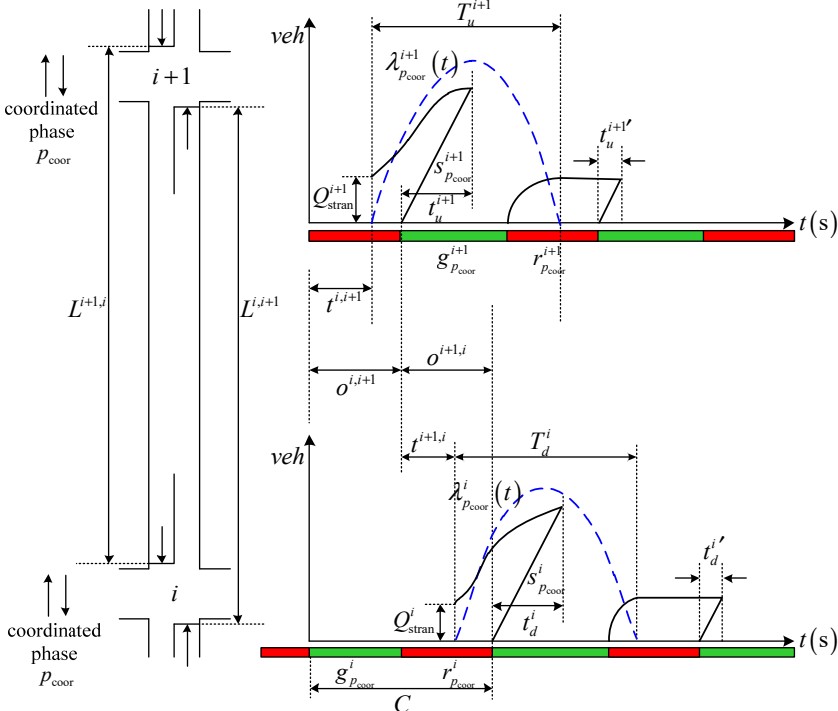

**Figure 1.** Schematic diagram of vehicle delay in coordinated direction when the offset is $o^{i,i+1}$.

(1)   Delay before and during the green time

①  Intersection $i+1$ (upstream direction):

$$d_u^{i+1} = \int_{t^{i,i+1}}^{o^{i,i+1}+t_u^{i+1}} \left[ Q_{\text{stran}}^{i+1} + \int_{t^{i,i+1}}^{t} \lambda_{p_{\text{coor}}}^{i+1}(t)dt \right] dt - 0.5 \cdot t_u^{i+1} \cdot s_{p_{\text{coor}}}^{i+1} \cdot t_u^{i+1} \tag{8}$$

where $t_u^{i+1}$ is the time required for queuing vehicles to dissipate during the green time at intersection $i + 1$, which should be satisfied with

$$Q_{\text{stran}}^{i+1} + \int_{t^{ii+1}}^{o^{i,i+1}+t_u^{i+1}} \lambda_{p_{\text{coor}}}^{i+1}(t)dt - s_{p_{\text{coor}}}^{i+1} \cdot t_u^{i+1} = 0 \tag{9}$$

②  Intersection $i$ (downstream direction):

$$d_d^i = \int_{o^{i,i+1}+t^{i+1,i}}^{C+t_d^i} \left[ Q_{\text{stran}}^i + \int_{o^{i,i+1}+t^{i+1,i}}^{t} \lambda_{p_{\text{coor}}}^i(t)dt \right] dt - 0.5 \cdot t_d^i \cdot s_{p_{\text{coor}}}^i \cdot t_d^i \tag{10}$$

where $t_d^i$ is the time required for queuing vehicles to dissipate during the green time at intersection $i$, which should be satisfied with

$$Q_{\text{stran}}^i + \int_{o^{i,i+1}+t^{i+1,i}}^{C+t_d^i} \lambda_{p_{\text{coor}}}^i(t)dt - s_{p_{\text{coor}}}^i \cdot t_d^i = 0 \tag{11}$$

(2)   Delay during the red time

①  Intersection $i+1$ (upstream direction):

$$d_u^{i+1\prime} = \int_{o^{i,i+1}+g_{p_{\text{coor}}}^{i+1}}^{t^{i,i+1}+T_u^{i+1}} \left[ \int_{o^{i,i+1}+g_{p_{\text{coor}}}^{i+1}}^{t} \lambda_{p_{\text{coor}}}^{i+1}(t)dt \right] dt + \left( o^{i,i+1} + C - t^{i,i+1} - T_u^{i+1} \right) Q_u^{i+1} + 0.5 t_u^{i+1\prime} Q_u^{i+1} \tag{12}$$

where $Q_u^{i+1} = \int_{o^{i,i+1}+g_{p_{\text{coor}}}^{i+1}}^{t^{i,i+1}+T_u^{i+1}} \lambda_{p_{\text{coor}}}^{i+1}(t)dt$, $t_u^{i+1\prime} = \frac{Q_u^{i+1}}{s_{p_{\text{coor}}}^{i+1}}$.

②  Intersection $i$ (downstream direction):

$$d_d^{i\prime} = \int_{C+g_{p_{\text{coor}}}^i}^{o^{i,i+1}+t^{i+1,i}+T_d^i} \left[ \int_{C+g_{p_{\text{coor}}}^i}^{t} \lambda_{p_{\text{coor}}}^i(t)dt \right] dt + \left( 2C - o^{i,i+1} - t^{i+1,i} - T_d^i \right) Q_d^i + 0.5 t_d^{i\prime} Q_d^i \tag{13}$$

where $Q_d^i = \int_{C+g_{p_{\text{coor}}}^i}^{o^{i,i+1}+t^{i+1,i}+T_d^i} \lambda_{p_{\text{coor}}}^i(t)dt$, $t_d^{i\prime} = \frac{Q_d^i}{s_{p_{\text{coor}}}^i}$.

(3)   Delay of the coordinated direction under the two-way arterial signal coordination

$$D = d_u^{i+1} + d_u^{i+1\prime} + d_d^i + d_d^{i\prime} \tag{14}$$

Therefore, the two-way arterial signal coordination control model [21] (M.1) is established as

$$\min_{o^{1,2}, o^{2,3}, \cdots, o^{n-1,n}} (D) = \min_{o^{1,2}, o^{2,3}, \cdots, o^{n-1,n}} \left[ \sum_{i=1}^{n} \left( d_u^{i+1} + d_u^{i+1\prime} + d_d^i + d_d^{i\prime} \right) \right] \tag{15}$$

subject to (5)–(13).

By solving this model, the optimal offset $o^{1,2}, o^{2,3}, \cdots, o^{n-1,n}$ for two-way arterial signal coordination with minimum vehicle delay in the coordinated direction can be obtained.

Model M.1 shows the relationship between the offset and the vehicle delay in the coordinated direction. According to this model, the basic conditions for determining the optimal offset are as follows: (1) the signal timing parameters at each intersection of

urban arterial; (2) the intersection is undersaturated at the urban arterial. However, this model does not provide a computational method for timing intersection signals under time-varying traffic flow conditions. If fixed-time control is used at each intersection, the effectiveness of arterial signal coordination will be limited. In order to improve the efficiency of arterial signal coordination, it is necessary to adapt the signal timing parameters of each intersection to time-varying traffic flow. In response to this issue, this paper adopts the reverse causal-effect modeling approach to establish a real-time optimization control model for arterial signal coordination.

### 2.2. Two-Way Arterial Signal Coordination Model Based on Reverse Causal-Effect Modeling Approach

2.2.1. Intersection Signal Timing Model Based on Reverse Causal-Effect Modeling Approach

Liu et al. established a signal optimization dynamic linear programming model using a reversed causal-effect approach [22]. The signalized intersection is regarded as a normal highway bottleneck. Both traffic arrivals and departures are modeled by smooth continuous functions of time, as if there were no interruptions to traffic flows from signals. The idea of the reverse causal-effect approach is to first optimize departure flow rate based on saturation flow rate and arrival rate of each approach at intersections and then convert the optimal departure flow rate into the intersection signal timing parameters.

Through the analysis of this model, it can be seen that this model allocates departure flow based on the importance of each approach at the intersection (i.e., the magnitude of saturation flow rate) and then converts it into signal timing parameters. However, for unsaturated intersections, this model cannot be applicable. Because the arrival rate is less than the saturated flow rate, the intersection cannot be regarded as a traffic bottleneck. In order to calculate the signal timing at an unsaturated intersection, the model needs to be modified by reducing the saturated flow rate with a certain proportion. In this way, a traffic bottleneck can be artificially produced when it meets an unsaturated intersection during the process of the modeling.

The specific reduction method of saturated flow rate is to multiply the saturated flow rate of each approach of intersection by the reduction coefficient $\phi^i$:

$$\phi^i(t) = \begin{cases} 1 & , \ \max_p\left[\lambda_p^i(t)\right] > s_p^i \\ \dfrac{\max_p\left[\lambda_p^i(t)\right]}{s_p^i}(1+\alpha), & \max_p\left[\lambda_p^i(t)\right] \leq s_p^i \end{cases} \tag{16}$$

$$s_{p_{\mathrm{narr}}}^i = s_p^i \cdot \phi^i \tag{17}$$

where $\alpha \in (0,1)$ is adjustment factor.

When the intersection is unsaturated, the saturated flow of each approach of the intersection is reduced proportionally by Formula (17); then, the bottleneck congestion occurs when the traffic flow passes through the intersection. In this way, Liu's model can be used to calculate the signal timing scheme of unsaturated intersections.

In order to ensure that the vehicles in the coordinated phase are emptied as much as possible, the concept of phase clearance reliability (PCR) [23] is introduced in this paper to ensure that the vehicle emptying reliability in the coordinated phase meets a certain level.

$$P_r\left[\eta_{p_{\mathrm{coor}}}^i(t_k)s_{p_{\mathrm{coor}}}^i \geq \lambda_{p_{\mathrm{coor}}}^i(t_k) + Q_{p_{\mathrm{coor}}}^i(t_k)/dt\right] \geq \theta_{p_{\mathrm{coor}}}^i, \qquad \forall i \tag{18}$$

The formula above indicates the probability that the coordinated phase green signal ratio is greater than or equal to the actual required green signal ratio and is not less than $\alpha^{p_b} \in [0,1]$. (For example, if $\alpha^{p_b} = 1$, it means that the PCR value of the coordinated phase is 100%, which means that the coordinated phase queue is completely emptied; If $\alpha^{p_b} = 0.95$, it means that the PCR value of the coordinated phase is 95%; that is, the coordinated phase reserves 5% green signal ratio adjustment space for other phases.)

Thus, by modifying Liu's model through the Formulas (16) and (17) and adding the constraint condition Formula (18), the signal timing calculation model (M.2) with the constraint of the coordinated phase clearance reliability can be obtained for the undersaturated intersection.

2.2.2. Offset Real-Time Optimization Model Based on Reverse Causal-Effect Modeling Approach

In this paper, a two-stage real-time arterial signal coordination control model (M.3) is established based on models M.1 and M.2. The basic ideas of the model are as follows: firstly, the model M.2 is used to calculate the real-time signal timing scheme of each intersection. Then, the model M.1 is used to calculate the optimal offset of arterial coordinated control based on the real-time signal timing scheme.

The first stage: Use model M.2 to calculate the real-time signal timing scheme of each intersection.

$$\min F = \sum_{p \in P} \sum_{m \in M^p} \int_{t_k} \left\{ \int_u \lambda_{p_m}^i(u) du - \int_u \mu_{p_m}^i(u) du \right\} dt \tag{19}$$

$$s.t. \quad \sum_{p \in P} \eta_p^i(t_k) \leq \eta^i \tag{20}$$

$$\eta_{p,\min}^i \leq \eta_p^i(t_k) \leq \eta_{p,\max}^i, \qquad \forall p \in P \tag{21}$$

$$\mu_{p_m}^i(t_k) \leq \eta_{p_m}^i(t_k) s_p^i, \qquad \forall m \in M^p \tag{22}$$

$$\mu_{p_m}^i(t_k) \leq Q_{p_m}^i(t_{k-1})/dt + \lambda_{p_m}^i(t_k), \qquad \forall m \in M^p \tag{23}$$

$$\mu_{p_m}^i(t_k) \geq 0, \quad \eta_p^i(t_k) \geq 0, \quad \forall p \in P, \, m \in M^p \tag{24}$$

$$\eta_p^i(t_k) = \max_m \left[ \frac{\sum\limits_{m \in M} \mu_{p_m}^i(t_k)}{\sum\limits_{m \in M} s_p^i} \right] \tag{25}$$

$$Q_{p_m}^i(t_{k-1}) = \int_{t_k} \left\{ \lambda_{p_m}^i(t_{k-1}) - \mu_{p_m}^i(t_{k-1}) \right\} dt \tag{26}$$

$$P_r \left[ \eta_{p_{\mathrm{coor}}}^i(t_k) s_{p_{\mathrm{coor}}}^i \geq \lambda_{p_{\mathrm{coor}}}^i(t_k) + Q_{p_{\mathrm{coor}}}^i(t_k)/dt \right] \geq \theta_{p_{\mathrm{coor}}}^i, \qquad \forall i \tag{27}$$

$$s_{p_{\mathrm{narr}}}^i = s_p^i \cdot \phi^i \tag{28}$$

The second stage: Use model M.1 to calculate the offset of two-way arterial signal coordination.

Objective function: (15) is subject to (5)–(13).

*2.3. Model Solving*

2.3.1. Model Simplification

In the actual traffic flow scene, it is difficult to make the traffic flow arrival rate accurate to the second. In order to facilitate the model calculation, we assume vehicles uniformly arrive at intersections within each cycle, though the arrival rates could vary from cycle to cycle. This not only ensures the real-time adaptability of the model to the traffic flow, but also simplifies the model.

(1) The first stage model (M.2) simplification

Since $\lambda^i_{p_m}$ and $C$ are uniform within each cycle, objective function can be written as

$$
\begin{aligned}
\min F &= \sum_{p \in P} \sum_{m \in M^p} \int_{t_k} \left\{ \int_u \lambda^i_{p_m}(u) du - \int_u \mu^i_{p_m}(u) du \right\} dt \\
&= \sum_{p \in P} \sum_{m \in M^p} \left[ \lambda^i_{p_m}(k) - \mu^i_{p_m}(k) \right] C
\end{aligned}
\tag{29}
$$

Therefore, the objective function Formula (19) is converted to

$$
\max \sum_{p \in P} \sum_{m \in M^p} \mu^i_{p_m}(k)
\tag{30}
$$

Constraint Formula (23) is converted to

$$
\mu^i_{p_m}(k) \leq Q^i_{p_m}(k-1)/dt + \lambda^i_{p_m}(k), \qquad \forall m \in M^p
\tag{31}
$$

Constraint Formula (26) is converted to

$$
Q^i_{p_m}(k) = Q^i_{p_m}(k-1) + \lambda^i_{p_m}(k-1)C - \mu^i_{p_m}(k-1)C, \qquad \forall p \in P
\tag{32}
$$

Other constraints of the model remain unchanged.

(2) The second stage model (M.1) simplification

Since $\lambda^i_{p_m}$ is assumed to be constant in the one cycle, the delay formula in model M.1 can be simplified, and the simplification process is shown in Appendix A.

$$
d^{i+1}_u = A^{i+1}_u \left( o^{i,i+1} - t^{i,i+1} \right)^2 + B^{i+1}_u \left( o^{i,i+1} - t^{i,i+1} \right) + C^{i+1}_u
\tag{33}
$$

$$
d^i_d = A^i_d \left( o^{i+1,i} - t^{i+1,i} \right)^2 + B^i_d \left( o^{i+1,i} - t^{i+1,i} \right) + C^i_d
\tag{34}
$$

$$
d^{i+1\prime}_u = A^{i+1\prime}_u \left( o^{i,i+1} - t^{i,i+1} \right)^2 + B^{i+1\prime}_u \left( o^{i,i+1} - t^{i,i+1} \right) + C^{i+1\prime}_u
\tag{35}
$$

$$
d^{i\prime}_d = A^{i\prime}_d \left( C - o^{i,i+1} - t^{i+1,i} \right)^2 + B^{i\prime}_d \left( C - o^{i,i+1} - t^{i+1,i} \right) + C^{i\prime}_d
\tag{36}
$$

where $A$, $B$ and $C$ are the correlation coefficients of the delay calculation, as detailed in Appendix A.

Therefore, when the offset is $o^{i,i+1}$, the total delay $d^{i,i+1}$ in the coordinated direction can be expressed as

$$
\begin{aligned}
d^{i+1} &= d^{i+1}_u + d^{i+1\prime}_u \\
&= \left( A^{i+1}_u + A^{i+1\prime}_u \right) \left( o^{i,i+1} - t^{i,i+1} \right)^2 + \left( B^{i+1}_u + B^{i+1\prime}_u \right) \left( o^{i,i+1} - t^{i,i+1} \right) + \left( C^{i+1}_u + C^{i+1\prime}_u \right)
\end{aligned}
\tag{37}
$$

$$
\begin{aligned}
d^i &= d^i_d + d^{i\prime}_d \\
&= \left( A^i_d + A^{i\prime}_d \right) \left( o^{i+1,i} - t^{i+1,i} \right)^2 + \left( B^i_d + B^{i\prime}_d \right) \left( o^{i+1,i} - t^{i+1,i} \right) + \left( C^i_d + C^{i\prime}_d \right) \\
&= \left( A^i_d + A^{i\prime}_d \right) \left( C - o^{i,i+1} - t^{i+1,i} \right)^2 + \left( B^i_d + B^{i\prime}_d \right) \left( C - o^{i,i+1} - t^{i+1,i} \right) + \left( C^i_d + C^{i\prime}_d \right)
\end{aligned}
\tag{38}
$$

$$
d^{i,i+1} = d^{i+1} + d^i
\tag{39}
$$

Find the first partial derivative of $o^{i,i+1}$ for the total delay $d^{i,i+1}$ of adjacent intersections $i, i+1$, and make it 0; that is

$$
\begin{aligned}
\frac{\partial d^{i,i+1}}{\partial o^{i,i+1}} &= \frac{\partial\left(d^{i+1}+d^i\right)}{\partial o^{i,i+1}} = 2\left(A_u^{i+1}+A_u^{i+1\prime}\right)\left(o^{i,i+1}-t^{i,i+1}\right)-2\left(A_d^i+A_d^{i\prime}\right)\left(C-o^{i,i+1}-t^{i+1,i}\right) \\
&\quad +\left(B_u^{i+1}+B_u^{i+1\prime}\right)-\left(B_d^i+B_d^{i\prime}\right)=0
\end{aligned}
\tag{40}
$$

From the Formula (40), $o^{i,i+1}$ can be obtained as

$$
o^{i,i+1}=\frac{2\left(A_d^i+A_d^{i\prime}\right)\left(C-t^{i+1,i}\right)+2\left(A_u^{i+1}+A_u^{i+1\prime}\right)t^{i,i+1}+\left(B_u^{i+1}+B_u^{i+1\prime}\right)-\left(B_d^i+B_d^{i\prime}\right)}{2\left(A_u^{i+1}+A_u^{i+1\prime}+A_d^i+A_d^{i\prime}\right)}
\tag{41}
$$

From the Formula (41), it can be seen that the optimal offset of two-way signal coordination at adjacent intersections is dependent on the arrival rate, signal timing, saturation flow rate, and $t^{i,i+1}$, and the optimal offset at adjacent intersections can be calculated by these conditions.

Through the above analysis, the calculation formula of vehicle delay in the coordinated direction of the traffic trunk line is as follows:

$$
D=\sum_{i=1}^n d^{i,i+1}=\sum_{i=1}^n\left(d^{i+1}+d^i\right)
\tag{42}
$$

Assuming that there are n intersections in the city traffic trunk line, there are corresponding n-1 offsets $o^{1,2}, o^{2,3}, \cdots, o^{n-1,n}$ for signal coordination. According to Formula (41), the offset between adjacent intersections only depends on the traffic flow, signal timing, saturation flow, and $t^{i,i+1}$ and is not related to other parameters. Therefore, the offset of the traffic trunk line can be calculated one by one by using Formula (41); that is, all the offset in the coordinated direction can be solved in turn from the first intersection of the traffic trunk line.

### 2.3.2. Solution Algorithm

Model M.3 is a two-stage optimization model, where the first stage aims to solve for the signal timing at each intersection, and the second stage aims to solve for the optimal offset for arterial signal coordination.

The first stage model is a single intersection signal timing optimization model, which is a linear programming model and can be solved quickly and accurately. On this basis, this paper further considers the optimization of arterial signal coordination and adds the constraint condition of coordinate phase priority (Formula (27)), which means ensuring that the PCR value of the coordinated phase meets a certain level. However, this constraint makes the original model M.2 become a nonlinear programming model. If the constraint condition (27) is removed for the model M.2, the model M.2 will degenerate into the original linear programming model.

Therefore, the basic idea of the first-stage model solving algorithm is as follows: firstly, the green time of each intersection is allocated for the first time by using the model M.2. Then, whether the green time of the coordinated phase satisfies the constraint condition (27) will be judged. If it is satisfied, the signal timing parameters of each intersection are obtained, and the second-stage model solves the optimal offset; if it is not satisfied, the green time is adjusted according to the PCR value of the coordinated phase. The adjustment method is as follows: adding a unit adjustment amount $\Delta\lambda_{\text{coor}}$ to the arrival rate $\lambda_{p_{\text{coor}}}^i(t)$ of the coordinated phase and recalculating the green time of each phase; then, the green time of the non-coordinated phase will be adjusted to the coordinated phase. This step is repeated until the PCR value of the coordinated phase satisfies the constraint condition. Then, one can turn to the second stage model to solve the optimal offset of the arterial signal coordination.

The second stage model M.1 can be solved for the optimal offset at each intersection of the arterial signal coordination using Equation (41) in a sequential recursive manner based on the first stage model. The algorithm flow chart is shown in Figure 2.

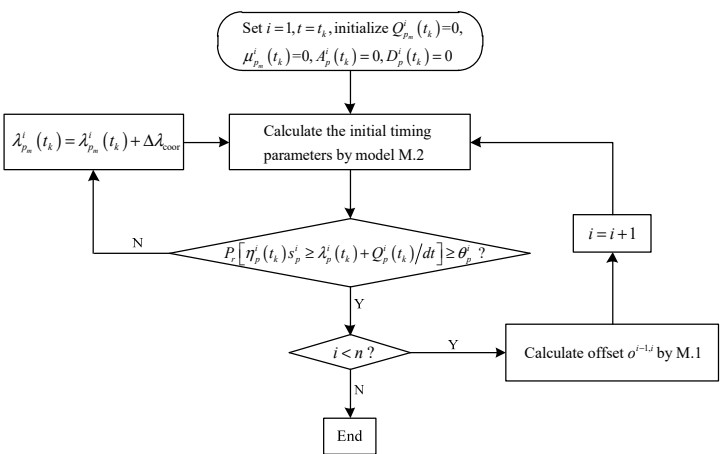

**Figure 2.** Algorithm flow chart.

Based on the aforementioned analysis, it is evident that the simplified model M.2 becomes a linear programming model after determining the PCR value. Similarly, model M.1 is also simplified into a linear model. Consequently, the model M.3 is a two-stage linear programming model when considering the determination of PCR. Therefore, Model M.3 can be easily solved.

## 3. Case Study

As shown in Figure 3, the basic layout, saturation flow rate, and phase structure of an example arterial road. Other parameter settings are shown in Table 2. This study employs an arterial road consisting of three intersections to illustrate the applicability of the proposed model, and five traffic flow scenarios are designed for each intersection (as shown in Table 3).

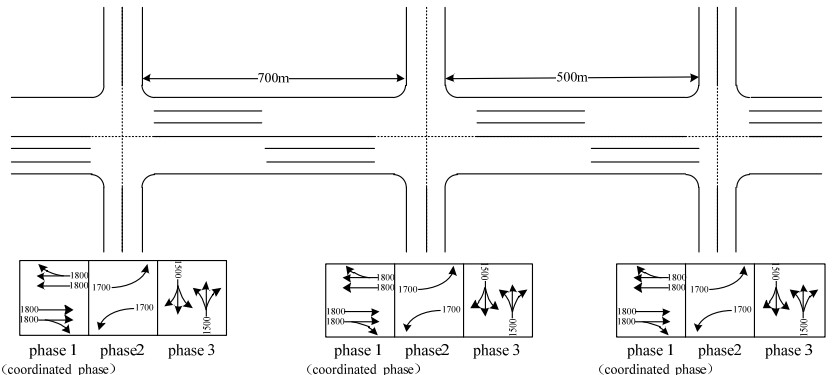

**Figure 3.** Intersection layouts, saturation flow rates (pcu/h), and phase structures.

**Table 2.** The simulation parameters.

| Parameter | Value |
|---|---|
| Maximum green time | $g^i_{1max} = 65$ s $\quad g^i_{2max} = 30$ s $\quad g^i_{3max} = 30$ s |
| Minimum green time | $g^i_{1min} = 35$ s $\quad g^i_{2min} = 13$ s $\quad g^i_{3min} = 10$ s |
| Inter-green time | 4 s (including 3 s amber and 1 s all red) |
| Average vehicle speed | 35 km/h |

**Table 3.** The dynamic traffic demand.

| Traffic Flow Scenario | | West Bound | | | North Bound | East Bound | | | South Bound |
|---|---|---|---|---|---|---|---|---|---|
| | | ↗ | → | ⇥ | ⋀ | ⌒ | ← | ⬳ | ⋎ |
| Intersection 1 | 1 | 150 | 430 | 430 | 130 | 110 | 400 | 400 | 120 |
| | 2 | 160 | 530 | 530 | 150 | 120 | 480 | 480 | 140 |
| | 3 | 170 | 630 | 630 | 170 | 130 | 560 | 560 | 160 |
| | 4 | 180 | 730 | 730 | 190 | 140 | 640 | 640 | 180 |
| | 5 | 190 | 830 | 830 | 210 | 150 | 720 | 720 | 160 |
| Intersection 2 | 1 | 200 | 620 | 620 | 150 | 150 | 550 | 550 | 140 |
| | 2 | 210 | 720 | 720 | 160 | 160 | 630 | 630 | 150 |
| | 3 | 220 | 820 | 820 | 170 | 170 | 710 | 710 | 160 |
| | 4 | 230 | 920 | 920 | 180 | 180 | 790 | 790 | 170 |
| | 5 | 240 | 1020 | 1020 | 190 | 190 | 870 | 870 | 180 |
| Intersection 3 | 1 | 110 | 420 | 420 | 120 | 120 | 370 | 370 | 100 |
| | 2 | 170 | 520 | 520 | 130 | 130 | 450 | 450 | 110 |
| | 3 | 180 | 620 | 620 | 140 | 140 | 530 | 530 | 120 |
| | 4 | 190 | 720 | 720 | 150 | 150 | 610 | 610 | 130 |
| | 5 | 200 | 820 | 820 | 160 | 160 | 690 | 690 | 140 |

We choose Allsop's method, Webster's method, and the M.2 method to calculate the basic timing scheme. Allsop's method consists of three steps: estimating traffic volume, distributing traffic flow and calculating travel time, and determining the best signal timing scheme. The core idea of this method is to find the most reasonable signal timing scheme by considering the traffic flow, road network, and signal period. Webster's method aims at minimizing vehicle delays at intersections, calculates the optimal cycle time, and allocates the green light time according to the traffic flow. The core idea of this model is to minimize the stopping time on the premise of ensuring traffic safety.

The offset of adjacent intersections is calculated by the M.1 model to check their solution qualities in terms of the average delay and average overflow. The signal timing scheme and offset calculation results of each intersection are shown in Table 4.

We select intersection 2 as an example to verify the performance of coordinated two-stage control logic and coordinated actuated control logic in the two indicators of average delay and average overflow. The basic parameters in the actuated signal control of each intersection are maximum green time $g^i_{1max} = 60$ s, $g^i_{2max} = 30$ s, $g^i_{3max} = 30$ s and minimum green time $g^i_{1min} = 35$ s, $g^i_{2min} = 13$ s, $g^i_{3min} = 10$ s; the unit green extension is 2 s. The average approach delay and the average overflow under the five traffic flow scenarios are shown in Tables 5 and 6.

First of all, from Table 5, we can see the performance of the average delay at intersection 2 under two coordinated control logics, among which the coordinated control logic based on model M.2 has the smallest delay at intersection 2. It can be further seen from Figure 4 that its performance is more significant under high traffic demands. The coordinated control logic based on Allsop's and Webster's methods performs better at low traffic demands, but the average vehicle delay increases obviously under the high traffic demand. This is because the two-stage arterial signal coordination model based on the M.2 method eliminates the overflow of the coordinated phase while minimizing the total intersection delay, which reduces the average delay at intersections significantly.

**Table 4.** Green time and offset of each intersection.

| Model | Traffic Flow Scenario | Cycle | Intersection 1 | | | Intersection 2 | | | Intersection 3 | | | Offset (M.1) | Calculation Time |
|---|---|---|---|---|---|---|---|---|---|---|---|---|---|
| | | | P1 | P2 | P3 | P1 | P2 | P3 | P1 | P2 | P3 | | |
| M.2 | 1 | 70 | 33 | 18 | 12 | 38 | 14 | 11 | 34 | 17 | 12 | $o^{1,2} = 44$ s, $o^{2,3} = 48$ s | 1.36 s |
| | 2 | 70 | 35 | 17 | 11 | 40 | 13 | 10 | 35 | 17 | 11 | $o^{1,2} = 44$ s, $o^{2,3} = 45$ s | |
| | 3 | 90 | 47 | 17 | 16 | 51 | 16 | 13 | 47 | 19 | 14 | $o^{1,2} = 35$ s, $o^{2,3} = 51$ s | |
| | 4 | 90 | 51 | 18 | 11 | 56 | 13 | 11 | 50 | 19 | 11 | $o^{1,2} = 37$ s, $o^{2,3} = 49$ s | |
| | 5 | 100 | 58 | 19 | 13 | 62 | 15 | 13 | 57 | 20 | 13 | $o^{1,2} = 42$ s, $o^{2,3} = 52$ s | |
| Allsop's | 1 | 70 | 31 | 14 | 13 | 32 | 16 | 12 | 32 | 16 | 12 | $o^{1,2} = 43$ s, $o^{2,3} = 49$ s | 2.31 s |
| | 2 | 70 | 35 | 13 | 11 | 35 | 15 | 11 | 34 | 14 | 12 | $o^{1,2} = 43$ s, $o^{2,3} = 46$ s | |
| | 3 | 90 | 43 | 20 | 16 | 44 | 21 | 16 | 43 | 21 | 15 | $o^{1,2} = 32$ s, $o^{2,3} = 52$ s | |
| | 4 | 90 | 44 | 21 | 15 | 45 | 20 | 15 | 45 | 20 | 15 | $o^{1,2} = 33$ s, $o^{2,3} = 51$ s | |
| | 5 | 100 | 48 | 22 | 19 | 50 | 23 | 18 | 47 | 23 | 20 | $o^{1,2} = 41$ s, $o^{2,3} = 53$ s | |
| Webster's | 1 | 70 | 32 | 15 | 13 | 34 | 14 | 12 | 33 | 15 | 12 | $o^{1,2} = 43$ s, $o^{2,3} = 41$ s | 1.73 s |
| | 2 | 70 | 34 | 16 | 10 | 35 | 14 | 11 | 34 | 15 | 11 | $o^{1,2} = 43$ s, $o^{2,3} = 45$ s | |
| | 3 | 90 | 43 | 20 | 16 | 44 | 21 | 16 | 42 | 21 | 17 | $o^{1,2} = 32$ s, $o^{2,3} = 52$ s | |
| | 4 | 90 | 45 | 22 | 13 | 46 | 21 | 13 | 44 | 22 | 14 | $o^{1,2} = 33$ s, $o^{2,3} = 50$ s | |
| | 5 | 100 | 50 | 23 | 18 | 53 | 22 | 17 | 49 | 24 | 17 | $o^{1,2} = 41$ s, $o^{2,3} = 52$ s | |

**Table 5.** Average approach and intersection delays under different traffic demand of intersection 2.

| Traffic Flow Scenario | Control Logic | Method | Phase 1 | | Phase 2 | | Phase 3 | | Avg. Intersection (s/veh) | Delay Different (%) |
|---|---|---|---|---|---|---|---|---|---|---|
| | | | East | West | East | West | North | South | | |
| 1 | coordinated two-stage | M.2 | 12 | 8 | 30 | 28 | 31 | 22 | 18.2 | -- |
| | | Allsop's | 13 | 9 | 31 | 32 | 27 | 19 | 19.6 | 7.69% |
| | | Webster's | 13 | 10 | 28 | 34 | 29 | 17 | 19.8 | 8.79% |
| | coordinated actuated | -- | 11 | 3 | 34 | 39 | 48 | 29 | 20.7 | 13.74% |
| 2 | coordinated two-stage | M.2 | 13 | 10 | 36 | 38 | 37 | 28 | 24.9 | -- |
| | | Allsop's | 15 | 12 | 53 | 62 | 71 | 37 | 28.1 | 12.85% |
| | | Webster's | 15 | 12 | 51 | 45 | 72 | 36 | 28.7 | 15.26% |
| | coordinated actuated | -- | 12 | 5 | 43 | 59 | 102 | 40 | 27.2 | 9.24% |
| 3 | coordinated two-stage | M.2 | 15 | 11 | 49 | 55 | 127 | 45 | 37.8 | -- |
| | | Allsop's | 18 | 13 | 87 | 109 | 119 | 57 | 42.7 | 12.96% |
| | | Webster's | 17 | 13 | 82 | 95 | 176 | 48 | 48.2 | 27.51% |
| | coordinated actuated | -- | 15 | 5 | 52 | 62 | 196 | 52 | 40.7 | 7.67% |
| 4 | coordinated two-stage | M.2 | 16 | 13 | 57 | 76 | 143 | 62 | 42.5 | -- |
| | | Allsop's | 19 | 15 | 101 | 138 | 137 | 61 | 51.3 | 20.71% |
| | | Webster's | 20 | 14 | 98 | 131 | 175 | 69 | 54.4 | 28.00% |
| | coordinated actuated | -- | 16 | 7 | 60 | 76 | 198 | 67 | 45.1 | 6.12% |
| 5 | coordinated two-stage | M.2 | 18 | 15 | 67 | 82 | 165 | 75 | 48.1 | -- |
| | | Allsop's | 22 | 16 | 113 | 152 | 162 | 79 | 59.2 | 23.08% |
| | | Webster's | 23 | 18 | 109 | 150 | 203 | 83 | 64.4 | 33.89% |
| | coordinated actuated | -- | 17 | 13 | 66 | 93 | 217 | 81 | 50.6 | 5.20% |

**Table 6.** Average approach and intersection overflows under different traffic demand of intersection 2.

| Traffic Flow Scenario | Control Logic | Method | Phase 1 | | Phase 2 | | Phase 3 | | Avg. Intersection Overflow |
|---|---|---|---|---|---|---|---|---|---|
| | | | East | West | East | West | North | South | |
| 1 | coordinated two-stage | M.2 | 0.01 | 0.01 | 1.2 | 0.5 | 0.9 | 0.3 | 0.6 |
| | | Allsop's | 0.05 | 0.06 | 2.2 | 1.6 | 1.7 | 0.5 | 1.1 |
| | | Webster's | 0.03 | 0.07 | 5.2 | 2.2 | 4.2 | 1.1 | 2.2 |
| | coordinated actuated | -- | 0.001 | 0.009 | 2.1 | 0.7 | 2.7 | 0.8 | 0.7 |
| 2 | coordinated two-stage | M.2 | 0.02 | 0.03 | 2.3 | 1.2 | 1.3 | 0.5 | 0.9 |
| | | Allsop's | 0.21 | 0.08 | 8.2 | 5.2 | 2.1 | 1.2 | 2.3 |
| | | Webster's | 0.15 | 0.1 | 11.3 | 7.2 | 7.3 | 2.3 | 4.6 |
| | coordinated actuated | -- | 0.002 | 0.028 | 2.8 | 0.9 | 4.2 | 0.9 | 1.1 |
| 3 | coordinated two-stage | M.2 | 0.03 | 0.05 | 3.2 | 2.1 | 2.6 | 0.6 | 1.3 |
| | | Allsop's | 0.3 | 0.4 | 17.5 | 10.3 | 9.5 | 2.4 | 5.8 |
| | | Webster's | 0.2 | 0.3 | 20.6 | 15.6 | 13.2 | 3.9 | 8.7 |
| | coordinated actuated | -- | 0.004 | 0.03 | 3.6 | 1.3 | 6.5 | 1.1 | 1.7 |
| 4 | coordinated two-stage | M.2 | 0.04 | 0.12 | 5.5 | 3.4 | 5.2 | 2.1 | 3.1 |
| | | Allsop's | 0.51 | 0.71 | 22.8 | 14.9 | 12.4 | 4.2 | 10.2 |
| | | Webster's | 0.29 | 0.49 | 29.6 | 20.9 | 19.1 | 7.2 | 13.4 |
| | coordinated actuated | -- | 0.006 | 0.1 | 6.2 | 1.6 | 9.2 | 1.6 | 3.6 |
| 5 | coordinated two-stage | M.2 | 0.09 | 0.2 | 7.3 | 6.8 | 8.2 | 3.6 | 5.3 |
| | | Allsop's | 0.7 | 0.9 | 32.3 | 21.3 | 17.6 | 6.8 | 14.3 |
| | | Webster's | 0.6 | 0.7 | 39.2 | 30.4 | 22.3 | 10.3 | 19.2 |
| | coordinated actuated | -- | 0.008 | 0.15 | 10.1 | 3.2 | 11.8 | 4.2 | 6.2 |

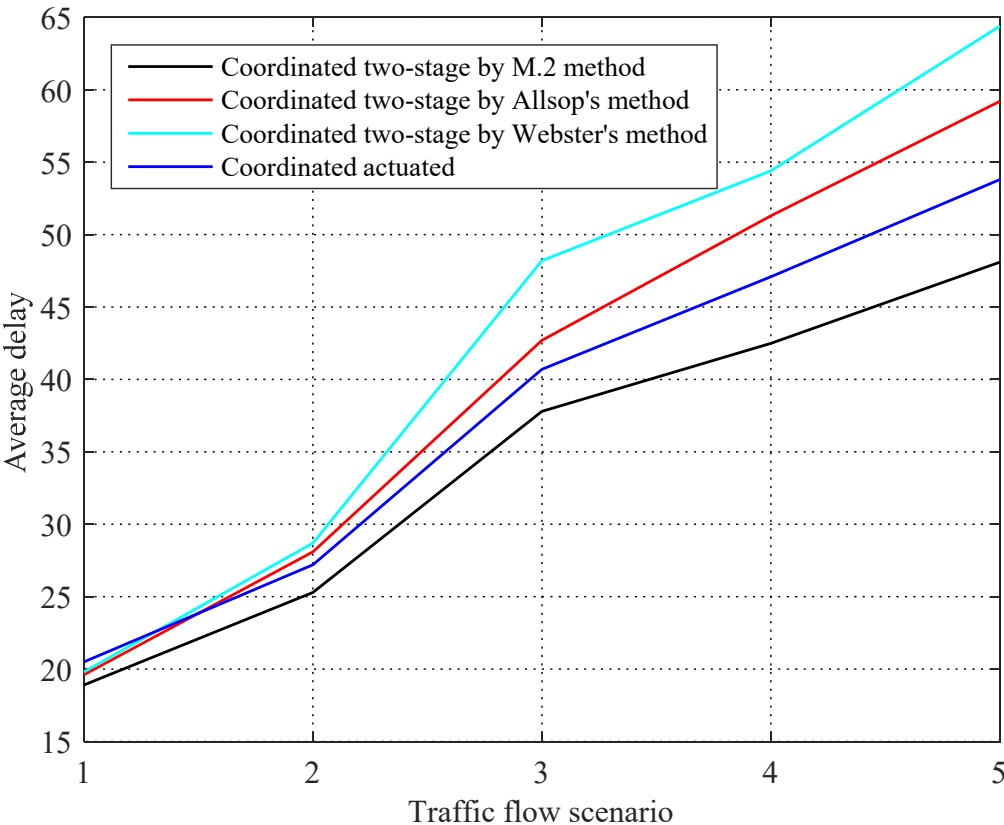

**Figure 4.** The average delay under different control schemes at intersection 2.

The coordinated phase under actuated control receives less delay than that under two-stage control, but there is a cost to increased delays in the non-coordinated phase. This is because the actuated coordination control ensures the priority of the coordinated phase but increases the delay of the other phases. Actuated control returns the green time of the non-coordinated phase to the coordinated phase, which empties the coordinated phase queue as much as possible but causes the increase of the vehicle delay of the non-coordinated phase. The average intersection delay under the actuated control is slightly higher than that under the two-stage control, which is mainly due to the inefficient utilization of green time.

The average approach overflow and the average intersection overflow of intersection 2 under the five traffic flow scenarios are shown in Table 6. Among the coordinated two-stage plans at high traffic demand (traffic Scenario 5), the M.2-based method generates the minimum amount of vehicle overflow, ranging from 0.09 to 8.2, with an average of 5.3. The number of overflow vehicles for the coordinated phase 1 under the M.2 two-stage coordinated signal control is the lowest among the five traffic flow scenarios. It clearly shows the advantages of the M.2-based method in overflow management of the coordinated phase. A similar trend is observed at the low traffic demand. The average overflow is reduced in the low traffic demand for all the three methods. Figure 5 further clearly shows that the M.2-based arterial coordination model can reduce the average overflow queue length at each intersection. Overflow comparison also demonstrates that actuated control highly prioritizes the coordinated phase by sacrificing the performance of the noncoordinated phases, and the overflow queue of the coordinated phase performs better than the other control models, but the overall average overflow queue of the intersection increases.

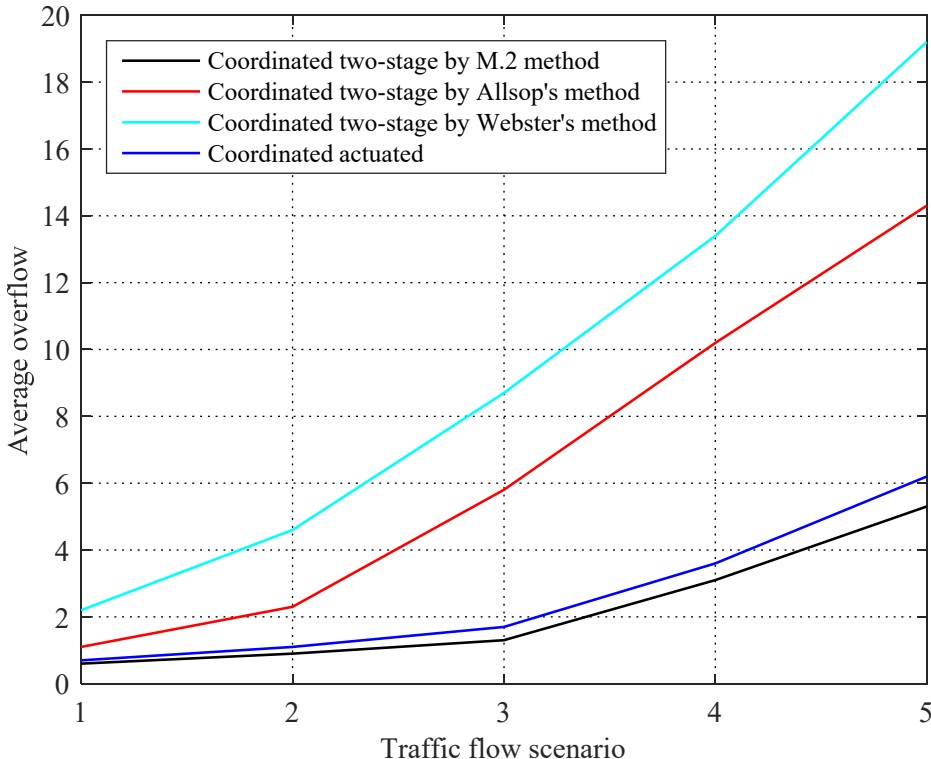

**Figure 5.** The average overflow under different control schemes at intersection 2.

In summary, the M.2-based two-stage arterial coordination model can effectively reduce the average delay and average overflow at intersections. This is because model M.2 adjusts for more green time in the coordinated phase (Table 4), and the main traffic flow at each intersection is distributed in the coordinated direction (Table 3). Model M.1 further optimizes the offset to ensure the priority of traffic flow at the coordinated phase, which reduces the overall delay and overflow queue at each intersection of the trunk line, thus improving the overall control benefit of the trunk line.

## 4. Conclusions

Arterial signal coordination can improve the efficiency of traffic flow in the coordinated direction, but it will also cause vehicle delay and queue length increases in the uncoordinated direction. In order to take the traffic operational efficiency of the coordinated and non-coordinated phases into consideration, a two-stage arterial signal coordinated control model under dynamic traffic demand is established in this paper. In the first stage, a calculation model for intersection signal timing based on phase clearing reliability is proposed by the reverse causal-effect modeling approach. In the second stage, an offset calculation model is established to achieve the goal of minimizing delay in the coordinated direction. Firstly, the concept of phase clearance reliability, which can be dynamically given according to the actual situation, is introduced in the model, increasing the flexibility of the model in practical use. Secondly, the model is based on the reverse causal-effect modeling approach, which can not only automatically identify the key traffic flow but also ensure the traffic operational efficiency in the coordinated direction and give full consideration to the traffic operational efficiency of each intersection. Thirdly, in order to solve the model conveniently, we simplify the model and design the corresponding algorithm. Finally, five dynamic traffic demands are designed to verify the applicability of the model. The results show that the two-stage arterial signal coordination model based on model M.2 can effectively reduce the average intersection delay and the average residual queue length compared with Webster's method and Allsop's method.

In this paper, we develop a two-stage real-time optimization model for arterial signal coordination based on the reverse causal-effect modeling approach by analyzing the relationship between vehicle delay and offset in the coordinated direction. This model can improve the traffic operational efficiency of the traffic trunk line, but the delay of the uncoordinated phase will increase under extreme traffic conditions. The future research direction is to consider how to extend this model to area traffic coordination control to improve the operation efficiency of area traffic control systems.

**Author Contributions:** Conceptualization, B.H. and B.L.; methodology, B.H.; software, Q.C.; validation, B.H., B.L. and Q.C.; formal analysis, B.H.; investigation, X.L.; resources, B.L.; data curation, B.H.; writing—original draft preparation, B.H.; writing—review and editing, B.H. and Q.C.; visualization, B.H.; supervision, B.L.; project administration, B.L.; funding acquisition, B.L. All authors have read and agreed to the published version of the manuscript.

**Funding:** This research was funded by the Natural Science Foundation of China (No. 52362044), "Double-First Class" Major Research Programs, Educational Department of Gansu Province (No. GSSYLXM-04), Colleges and universities Innovation Fund projects, Educational Department of Gansu Province (No. 2020B-108), and the Colleges and universities Scientific and Technological Research Projects, Educational Department of Gansu Province (No. 2022QB-060).

**Institutional Review Board Statement:** Not applicable.

**Informed Consent Statement:** Informed consent was obtained from all subjects involved in the study.

**Data Availability Statement:** Not applicable.

**Conflicts of Interest:** The authors declare no conflict of interest.

## Appendix A

Since $\lambda_{p_m}^i$ is assumed to be a fixed value in the same cycle, the cumulative curve of vehicles in one cycle is not a curve but a straight line (as shown in Figure A1), so the delay formula $d_u^{i+1}, d_u^{i+1}{}', d_d^i, d_d^i{}'$ in the model (14) needs to be modified.

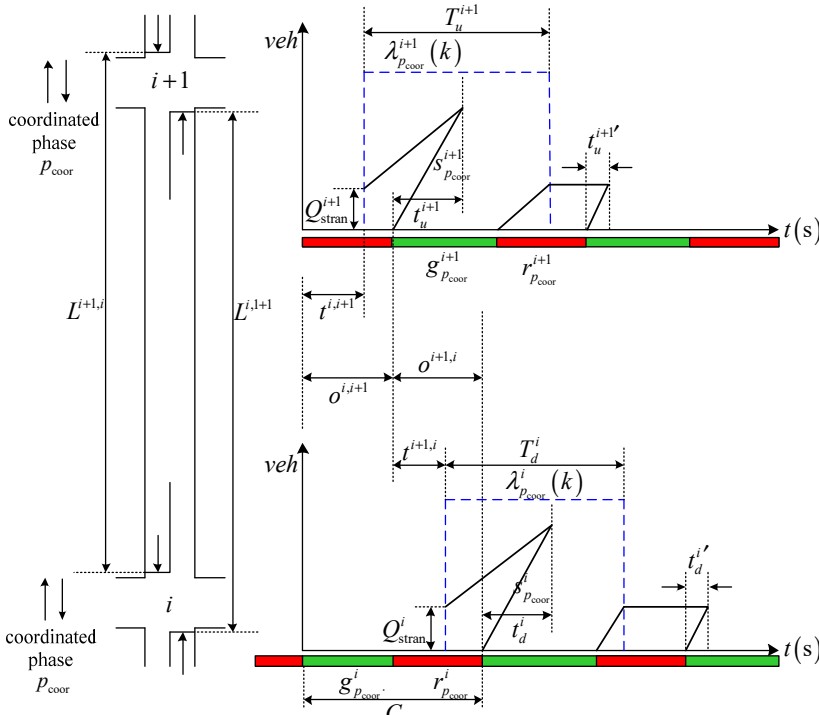

**Figure A1.** Schematic diagram of vehicle delay in coordinated direction when the offset is $o^{i,i+1}$.

(1) Delay before and during the green time

① Intersection $i+1$ (upstream direction):

$$
\begin{aligned}
d_u^{i+1} &= 0.5\left(o^{i,i+1} - t^{i,i+1} + t_u^{i+1}\right)\left(Q_{\text{stran}}^{i+1} + s_{p_{\text{coor}}}^{i+1} t_u^{i+1}\right) - 0.5 t_u^{i+1} s_{p_{\text{coor}}}^{i+1} t_u^{i+1} \\
&= 0.5\left[\left(o^{i,i+1} - t^{i,i+1} + t_u^{i+1}\right)Q_{\text{stran}}^{i+1} + \left(o^{i,i+1} - t^{i,i+1}\right)s_1^{i+1} t_u^{i+1}\right] \\
&= A_u^{i+1}\left(o^{i,i+1} - t^{i,i+1}\right)^2 + B_u^{i+1}\left(o^{i,i+1} - t^{i,i+1}\right) + C_u^{i+1}
\end{aligned}
\tag{A1}
$$

where $t_u^{i+1}$ is the queue discharge time during the green time at intersection $i$, which is calculated as

$$
t_u^{i+1} = \frac{Q_{\text{stran}}^{i+1} + \lambda_{p_{\text{coor}}}^{i+1}(k)\left(o^{i,i+1} - t^{i,i+1}\right)}{s_{p_{\text{coor}}}^{i+1} - \lambda_{p_{\text{coor}}}^{i+1}(k)}
\tag{A2}
$$

The coefficients in formula $d_u^{i+1}$ can be calculated as

$$
A_u^{i+1} = \frac{\lambda_{p_{\text{coor}}}^{i+1}(k)s_{p_{\text{coor}}}^{i+1}}{2\left(s_{p_{\text{coor}}}^{i+1} - \lambda_{p_{\text{coor}}}^{i+1}(k)\right)}
\tag{A3}
$$

$$
B_u^{i+1} = \frac{Q_{\text{stran}}^{i+1} s_{p_{\text{coor}}}^{i+1}}{s_{p_{\text{coor}}}^{i+1} - \lambda_{p_{\text{coor}}}^{i+1}(k)}
\tag{A4}
$$

$$
C_u^{i+1} = \frac{\left(Q_{\text{stran}}^{i+1}\right)^2}{2\left(s_{p_{\text{coor}}}^{i+1} - \lambda_{p_{\text{coor}}}^{i+1}(k)\right)}
\tag{A5}
$$

② Intersection $i$ (downstream direction)

Similarly, the delay before and after the green time at intersection i can be symmetrically obtained:

$$
d_d^i = A_d^i\left(o^{i+1,i} - t^{i+1,i}\right)^2 + B_d^i\left(o^{i+1,i} - t^{i+1,i}\right) + C_d^i
\tag{A6}
$$

where

$$
t_d^i = \frac{Q_{\text{stran}}^i + \lambda_{p_{\text{coor}}}^i(k)\left(o^{i+1,i} - t^{i+1,i}\right)}{s_{p_{\text{coor}}}^i - \lambda_{p_{\text{coor}}}^i(k)}
\tag{A7}
$$

$$
A_d^i = \frac{\lambda_{p_{\text{coor}}}^i(k)s_{p_{\text{coor}}}^i}{2\left(s_{p_{\text{coor}}}^i - \lambda_{p_{\text{coor}}}^i(k)\right)}
\tag{A8}
$$

$$
B_d^i = \frac{Q_{\text{stran}}^i s_{p_{\text{coor}}}^i}{s_{p_{\text{coor}}}^i - \lambda_{p_{\text{coor}}}^i(k)}
\tag{A9}
$$

$$
C_d^i = \frac{\left(Q_{\text{stran}}^i\right)^2}{2\left(s_{p_{\text{coor}}}^i - \lambda_{p_{\text{coor}}}^i(k)\right)}
\tag{A10}
$$

(2) Delay during red time

① Intersection $i+1$ (upstream direction)

The number of vehicles arriving at intersection i+1 after green time can be calculated as

$$
\begin{aligned}
Q_u^{i+1} &= \lambda_{p_{\text{coor}}}^{i+1}(k) \cdot \left[T_u^{i+1} - g_{p_{\text{coor}}}^{i+1} - \left(o^{i,i+1} - t^{i,i+1}\right)\right] \\
&= -\lambda_{p_{\text{coor}}}^{i+1}(k)\left(o^{i,i+1} - t^{i,i+1}\right) + \lambda_{p_{\text{coor}}}^{i+1}(k)\left(T_u^{i+1} - g_{p_{\text{coor}}}^{i+1}\right)
\end{aligned}
\tag{A11}
$$

The discharge time $t_u^{i+1\prime}$ of the number of vehicles arriving $Q_u^{i+1}$ can be calculated as

$$
t_u^{i+1\prime} = \frac{Q_u^{i+1}}{s_{p_{\text{coor}}}^{i+1}}
\tag{A12}
$$

Thus, the delay $d_u^{i+1\prime}$ caused by arriving vehicles during the red time at intersection $i + 1$ is

$$
\begin{aligned}
d_u^{i+1\prime} &= Q_u^{i+1} \cdot \left\{ r_u^{i+1} + \left[ C - T_u^{i+1} + \left( o^{i,i+1} - t^{i,i+1} \right) + t_u^{i+1\prime} \right] \right\} \\
&= 0.5 Q_u^{i+1} \left[ \frac{\left( s_{p_{\text{coor}}}^{i+1} - \lambda_{p_{\text{coor}}}^{i+1}(k) \right)}{s_{p_{\text{coor}}}^{i+1}} + \frac{s_{p_{\text{coor}}}^{i+1} \left( C - T_u^{i+1} + r_u^{i+1} \right)}{s_{p_{\text{coor}}}^{i+1}} \right] \\
&= A_u^{i+1\prime} \left( o^{i,i+1} - t^{i,i+1} \right)^2 + B_u^{i+1\prime} \left( o^{i,i+1} - t^{i,i+1} \right) + C_u^{i+1\prime}
\end{aligned}
\tag{A13}
$$

where

$$
A_u^{i+1\prime} = -\frac{\lambda_{p_{\text{coor}}}^{i+1}(k) \left[ s_{p_{\text{coor}}}^{i+1} - \lambda_{p_{\text{coor}}}^{i+1}(k) \right]}{2 s_{p_{\text{coor}}}^{i+1}}
\tag{A14}
$$

$$
B_u^{i+1\prime} = \frac{\lambda_{p_{\text{coor}}}^{i+1}(k) s_{p_{\text{coor}}}^{i+1} \left[ T_u^{i+1} - C \right] - \left[ \lambda_{p_{\text{coor}}}^{i+1}(k) \right]^2 \left( T_u^{i+1} - g_{p_{\text{coor}}}^{i+1} \right)}{s_{p_{\text{coor}}}^{i+1}}
\tag{A15}
$$

$$
C_u^{i+1\prime} = \frac{\lambda_{p_{\text{coor}}}^{i+1}(k) \left( T_u^{i+1} - g_{p_{\text{coor}}}^{i+1} \right) \left\{ s_{p_{\text{coor}}}^{i+1} \left( C - T_u^{i+1} + r_u^{i+1} \right) + \lambda_{p_{\text{coor}}}^{i+1}(k) \left[ T_u^{i+1} - g_{p_{\text{coor}}}^{i+1} \right] \right\}}{2 s_{p_{\text{coor}}}^{i+1}}
\tag{A16}
$$

② Intersection $i$ (downstream direction)

Similarly, the delay caused by vehicles arriving during the red time at intersection $i$ can be symmetrically obtained:

$$
\begin{aligned}
d_d^{i\prime} &= Q_d^i \cdot \left\{ r_d^i + \left[ C - T_d^i + \left( o^{i+1,i} - t^{i+1,i} \right) + t_d^{i\prime} \right] \right\} \\
&= 0.5 Q_d^i \left[ \frac{\left( s_{p_{\text{coor}}}^i - \lambda_{p_{\text{coor}}}^i(k) \right) \left( o^{i+1,i} - t^{i+1,i} \right)}{s_{p_{\text{coor}}}^i} + \frac{s_{p_{\text{coor}}}^i \left( C - T_d^i + r_d^i \right) + \lambda_{p_{\text{coor}}}^i(k) \left( T_d^i - g_{p_{\text{coor}}}^i \right)}{s_{p_{\text{coor}}}^i} \right] \\
&= A_d^{i\prime} \left( o^{i+1,i} - t^{i+1,i} \right)^2 + B_d^{i\prime} \left( o^{i+1,i} - t^{i+1,i} \right) + C_d^{i\prime} \\
&= A_d^{i\prime} \left( C - o^{i,i+1} - t^{i+1,i} \right)^2 + B_d^{i\prime} \left( C - o^{i,i+1} - t^{i+1,i} \right) + C_d^{i\prime}
\end{aligned}
\tag{A17}
$$

where

$$
Q_d^i = -\lambda_{p_{\text{coor}}}^i(k) \left( o^{i+1,i} - t^{i+1,i} \right) + \lambda_{p_{\text{coor}}}^i(k) \left( T_d^i - g_{p_{\text{coor}}}^i \right)
\tag{A18}
$$

$$
t_d^{i\prime} = \frac{Q_d^i}{s_{p_{\text{coor}}}^i}
\tag{A19}
$$

$$
A_d^{i\prime} = -\frac{\lambda_{p_{\text{coor}}}^i(k) \left[ s_{p_{\text{coor}}}^i - \lambda_{p_{\text{coor}}}^i(k) \right]}{2 s_{p_{\text{coor}}}^i}
\tag{A20}
$$

$$
B_d^{i\prime} = \frac{\lambda_{p_{\text{coor}}}^i(k) s_{p_{\text{coor}}}^i \left[ T_d^i - C \right] - \left[ \lambda_{p_{\text{coor}}}^i(k) \right]^2 \left( T_d^i - g_{p_{\text{coor}}}^i \right)}{s_{p_{\text{coor}}}^i}
\tag{A21}
$$

$$
C_d^{i\prime} = \frac{\lambda_{p_{\text{coor}}}^i(k) \left( T_d^i - g_{p_{\text{coor}}}^i \right) \left\{ s_{p_{\text{coor}}}^i \left( C - T_d^i + r_d^i \right) + \lambda_{p_{\text{coor}}}^i(k) \left[ T_d^i - g_{p_{\text{coor}}}^i \right] \right\}}{2 s_{p_{\text{coor}}}^i}
\tag{A22}
$$

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
