# Peer review of "A Two-Stage Real-Time Optimization Model of Arterial Signal Coordination Based on Reverse Causal-Effect Modeling Approach"

_sustainability, doi:10.3390/su151814035_

Round 1
Reviewer 1 Report
Aiming to minimize the expected intersection delay and the overflow of the coordinated direction, this work proposes a two-stage arterial signal coordination model. The topic is interesting, and the work is effective. However, the reviewer has some concerns as below. So I would recommend that before acceptance the manuscript should be rewritten in order to make it more accurate.
Some major problems are:
1. It is suggested to add the calculation time of each method.
2. The method of combining multiple methods is innovative to some extent. It is suggested to explain the Allsop’s and Webster’s methods in detail.
Author Response
Dear Reviewer,
Thank you for your time and consideration of our paper. We greatly appreciate the constructive criticisms of the reviewers. The responses to reviewers’comments are enclosed in the letter and the appropriate modifications have been made and highlighted in the manuscript. We also added complementary discussions and clarifications, and proofread the whole paper and made comprehensive improvements.
In addition to the following responses to the reviewers, we carefully went through the whole paper and made comprehensive improvements regarding the language, the style, the grammar, and the figures. Also, the focuses of the paper were better emphasized by adding complementary discussions and clarifications.
Should you have any questions, please feel free to contact me.
Sincerely,
Best regards,
Bin-bin Hao

Reviewer 2 Report
Please summarize the contributions in Introduction.
The algorithm should be presented in the algorithm form.
For the proposed algorithm, what about the running time?
Why not use color figures?
5.
Many typos need to be revised carefully. For example,
Line 20, what is “solution algorithm”?
Line 22, “Allsop’s method and Webster’s method” should be “the Allsop’s method and the Webster’s method”.
Line 32, “minimizing performance indicators method”, please make sure “indicators” or “indicator” in Line 47.
Line 40, “Zhang” should be “Zhang et. al.”.
Line 80, “Coordinated actuated signals can better adapt to fluctuating traffic flow better, and it is better in the case of small traffic flow”. I think you made a typo here.
Line 139, “(1)delay” should be “(1) delay”.
Line 145, “Where” should be “where”.
Author Response
Dear Reviewer,
Thank you for your time and consideration of our paper. We greatly appreciate the constructive criticisms of the reviewers. The responses to reviewers’comments are enclosed in the letter and the appropriate modifications have been made and highlighted in the manuscript. We also added complementary discussions and clarifications, and proofread the whole paper and made comprehensive improvements.
In addition to the following responses to the reviewers, we carefully went through the whole paper and made comprehensive improvements regarding the language, the style, the grammar, and the figures. Also, the focuses of the paper were better emphasized by adding complementary discussions and clarifications.
Please see the attachment.
Should you have any questions, please feel free to contact me.
Sincerely,
Best regards,
Bin-bin Hao

Reviewer 3 Report
To minimize the expected intersection delay and the overflow of the coordinated direction in an arterial road,the authors developed a two-stage real-time optimization model for arterial signal coordination based on the reverse causal-effect modeling approach. It is a very interesting study, while some questions should be clarified as follows.
1.The conclusions should be written in the abstract. And the abstract should be rewritten more logically.
2. expressed by
is more clear for reader to understand its meaning. The same as
and
3.Where the value of 0.5 in Eq.(8) is derived from?
4.In ,why are you sure that the vehicle cann’t pass the next intersection in one cycle. Otherwise, if the vehicle passes the next cross, the
would lose its meaning.
5. Do the two crosses in the coordinated direction have the same cycle time ? Are there any parameters to connect model M1 and M2?
6. The grammar errors should be checked carefully, such as, an arterial not a arterial , (15) subject to: (5)--(13) et al.
7. Is the algorithm only used for three-phase time setting crosses? If there is four-phase timing scheme, is the algorithm still suitable ?
8. The references are old, and some latest references in the near three years should be included.

The grammar errors should be checked carefully, such as, an arterial not a arterial , (15) subject to: (5)--(13) et al.
Author Response

(The authors gave the same response as above.)

Reviewer 4 Report
This paper presents a two-stage real-time optimization model for traffic signal coordination. The applied methodologies sound reasonable and the datasets seem to be sufficient. The proposed M.3 shows that a combined optimization of green time allocation and offsets will deliver a better and realistic result. Eventually, the finding may be useful for the further research works and for the practice. However, the comparisons of the references models and the conclusions are not clearly presented. Thus, some corrections and discussions must be done before a publication.
Some comments in details:
Line 78: “relatively single” => ”relatively simple”?.
Line 125-127: Symbol “t” not declared in the nomenclature list (Table 1)
Line 231-244: The first stage (M.2) corresponds to the consideration of uniform delay. However, it does not account for the platoon dispersion effect and thus the variability of speeds within the vehicle population. Please give a discussion here. The second stage (M.1) corresponds to the consideration of random delay. However, it does not account for the variability of random arrival flow rates. Please also give a discussion here.
Section 2.3.2: in my opinion, the deterministic case without randomness in the traffic flow, the optimization M.2 is equivalent to the M3. That is, the delay in the M.2 (first stage) will always have the same values and it is independent of offsets. In case of under saturation (degree of saturation x<1), PCR is always equal to 1 (total clearance for x<1). In my opinion, the parameter PCR is not suitable for evaluating the coordination. In case of a coordination, the queue in the coordinated phase should be avoided as good as possible. The best parameter for evaluation a coordination is the number of stops or the proportion of stopped vehicles in the coordinated phase. Here, the authors should give more discissions regarding the number of stops and coordination.
Tab.4 (also Tab.5): How is the delay obtained (calculated) for the scenarios Webster' and Allsop's? Which delay can be obtained if the Webster or HCM delay formula is used calculating delay for all scenarios?
Line 417-419: In my opinion, the comparison is not fair because the random delay is not accounted for in the M.3 (M.1) as it is the case in the Webster's or Allsop's model. However, Webster's and Allsop's model are developed only for isolated intersections. What will be the performance of the M.3 compared to the MAXBAND or Multiband model? Anyway, the proposed M.3 shows that a combined optimization of green time allocation and offsets will deliver a better and realistic result.
Author Response

(The authors gave the same response as above.)

Reviewer 5 Report
Dear Editor,
The proposed methodology do not justify the problem statement clearly. Major improvement is required to consider for publication.
The additional comments are included for the reviewed manuscript.
- The importance of problem statements are not well defined.
- The proposed method flow must be given using a flowchart.
- The simulation parameters must be listed in the form of a table for easy readability.
- It is suggested to keep recent references. Since almost all references are very old, which lacks the recent works.
- The results lack insights and are not presented properly.
- Since various phases are discussed in this work, the complexity analysis must be included.
Moderate editing of English language required.
Author Response

(The authors gave the same response as above.)

Round 2
Reviewer 2 Report
I am satisfied with this revision.
I am satisfied with this revision.
Reviewer 5 Report
The reviews are addressed.